# HALLUCINATION DETOX:
# SENSITIVE NEURON DROPOUT (SeND) FOR LARGE LANGUAGE MODEL TRAINING

Shahrad Mohammadzadeh[1, 4, *], Juan David Guerra[2, 4, *], Marco Bonizzato[2, 3, 4], Reihaneh Rabbany[1, 4], and Golnoosh Farnadi[1, 3, 4]

[1]McGill University, Montréal, Canada
[2]Polytechnique Montréal, Montréal, Canada
[3]Université de Montréal, Montréal, Canada
[4]Mila - Québec AI Institute, Montréal, Canada
{shahrad.mohammadzadeh, juan.guerra, marco.bonizzato,
reihaneh.rabbany, farnadig}@mila.quebec

## ABSTRACT

As large language models (LLMs) become increasingly deployed across various industries, concerns regarding their reliability, particularly due to hallucinations—outputs that are factually inaccurate or irrelevant to user input—have grown. Our research investigates the relationship between the training process and the emergence of hallucinations to address a key gap in existing research that focuses primarily on post hoc detection and mitigation strategies. Using models from the Pythia suite (70M–12B parameters) and several hallucination detection metrics, we analyze hallucination trends throughout training and explore LLM internal dynamics. We introduce **SEnsitive Neuron Dropout (SeND)**, a novel training protocol designed to mitigate hallucinations by reducing variance during training. SeND achieves this by deterministically dropping neurons with significant variability on a dataset, referred to as Sensitive Neurons. In addition, we develop an unsupervised hallucination detection metric, Efficient EigenScore (EES), which approximates the traditional EigenScore in 2x speed. This efficient metric is integrated into our protocol, allowing SeND to be both computationally scalable and effective at reducing hallucinations. Our empirical evaluation demonstrates that our approach improves LLM reliability at test time by up to 40% compared to normal training while also providing an efficient method to improve factual accuracy when adapting LLMs to domains such as Wikipedia and Medical datasets.

## 1 INTRODUCTION

### 1.1 MOTIVATION

In the era of increasingly advanced Large Language Models (LLMs), their widespread use across industries has raised concerns about reliability and safety, particularly due to errors and misuse. A key issue is hallucinations, where LLMs produce content misaligned with facts or user input (Huang et al., 2023a). Our research focuses on a specific type of hallucinations: confabulations, where LLMs generate inconsistent responses to inputs, switching between correct and incorrect information.

Previous research has largely focused on identifying and addressing hallucinations in large language models (LLMs), but the impact of the training process on hallucinations remains under-explored (Huang et al., 2023a; Rawte et al., 2023; Ye et al., 2023; Hong et al., 2024; Xu et al., 2024; Chen et al., 2024; Li et al., 2024; Gao et al., 2024b). This paper addresses this gap by investigating how the iterative learning process in LLMs leads to significant variance in hallucination behavior

---

*Equal Contribution

during training. This variability indicates that the model's factual confidence fluctuates, making it challenging to pinpoint a checkpoint at which the model has confidently learned facts.

As LLMs are deployed in high-risk industries, ensuring their reliability is crucial for user safety. However, this is not always achieved, leading to serious consequences, such as an Air Canada lawsuit over an LLM-generated incorrect policy (Garcia, 2024). Addressing such issues requires a deeper understanding of how hallucinations arise during training, enabling more reliable and efficient mitigation strategies beyond post-processing methods.

To explore these hallucination trends, we analyze models from 70 million to 12 billion parameters within Pythia suite (Biderman et al., 2023), assessing them across various training checkpoints and tasks. Our goal is to validate the oscillatory behavior observed in prior studies (Li et al., 2024) through evaluation metrics from HuggingFace and EleutherAI (Hong et al., 2024; Gao et al., 2024a), and to explore the correlation between model size, training progression, and hallucination patterns.

In response to the identified variance, we introduce a novel training protocol called **Sensitive Neuron Dropout** (SeND). SeND is designed to emphasize confident learning of facts, and in turn reduce the likelihood of confabulations, rather than solely minimizing the stochastic gradient descent (SGD) loss (e.g., cross-entropy). By selectively dropping Sensitive Neurons—those that exhibit significant fluctuations in contextual embeddings throughout training—SeND acts as a regularization technique that reduces hallucination variance and enhances the model's factual certainty. This provides a more reliable criterion for determining training termination, ensuring models not only achieve loss convergence but also display stable factual confidence. To maintain efficiency as model size and inference count increase, we propose the **Efficient EigenScore** (EES), an approximation metric for hallucination detection. EES replaces EigenScore (Chen et al., 2024), the primary metric used in our experiments, offering a scalable solution with high correlation to the original EigenScore.

Our contributions to the field can be summarized as follows, emphasizing that SeND enhances the training process but **does not replace** post-hoc solutions, which may still be required after training:[1]

1. Empirical verification of the **oscillatory nature of hallucinations in LLMs training** across various model scales and detection metrics.

2. **Sensitive Neuron Dropout (SeND)**, a training-time method designed to reduce hallucination variance and increase model factual confidence during training.

3. **Efficient EigenScore (EES)**, an efficient hallucination detection metric used to keep SeND efficient, achieving up to 2x speedup with minimal effects on accuracy.

## 1.2 RELATED WORK

The majority of research on hallucinations in language models has focused on detecting and mitigating this phenomenon rather than explaining its underlying causes. Recent techniques can be categorized into two main approaches: those that rely on output text or model probabilities at inference time (Manakul et al., 2023; Joshi et al., 2017; Li et al., 2023) and those that utilize internal representations or hidden layers of the model (Su et al., 2024; Chen et al., 2024; Kossen et al., 2024). While the former has demonstrated effectiveness, the latter offers deeper insights but often comes with computational trade-offs. Additionally, methods like Reinforcement Learning with Human Feedback (RLHF) have gained traction for enhancing model reliability (Yu et al., 2024). However, many of these post-hoc solutions enhance factual accuracy by layering algorithms atop pre-trained models, which can be inefficient. Our work addresses this gap by focusing on the internal dynamics of the model that contribute to hallucinations.

We use several metrics for evaluation, including Halueval (Li et al., 2023), FactScore (Min et al., 2023), SelfCheckGPT (Manakul et al., 2023), and XSum (Narayan et al., 2018), to validate our findings across different tasks. Given that the internal dynamics of the model have proven to be reliable candidates for assessing certainty and hallucination likelihood, we leverage methodologies such as EigenScore (Chen et al., 2024) and Semantic Entropy (Kossen et al., 2024), which detect hallucination risk by analyzing the variability in multiple high-temperature outputs. In our exper-

---

[1]For the code and datasets used, refer to our GitHub repository at: `https://anonymous.4open. science/r/SeND-Pythia/README.md`.

iments, we use the EigenScore metric alongside the HELM dataset created by Su et al. (2024) to detect hallucinations during training and in the development of SeND.

Regularization techniques have been introduced to fix the issue of variability, notably random neuron dropout, used to reduce the variance and ensure that no neuron is overpowering others (Srivastava et al., 2014; Baldi & Sadowski, 2013). Work such as that done by Santra et al. (2020); Ba & Frey (2013) aims to modify random neuron dropout to change the way neurons are dropped to a more deterministic, precise manner. This has allowed the authors to drop unimportant connections in a deep neural network to ensure that class discriminative information is propagated through the model correctly (Santra et al., 2020). Inspired by this, our aim is to target hallucinatory neurons in our models to ensure that factual information is propagated through.

A significant drawback of state-of-the-art detection techniques, particularly those relying on internal model dynamics, is their efficiency. Existing methods often necessitate multiple inferences and embedding generations, making the spectral analysis of embedding matrices computationally intensive and increasingly impractical as models and datasets grow (Chen et al., 2024; Su et al., 2024). To address these challenges, we propose the use of spectral theory for efficient approximation. This approach enables scalable hallucination detection while maintaining performance. By utilizing tools such as the Density of States (DOS) and the kernel polynomial method (KPM) for approximating EigenScore (Huang et al., 2023b; Lin et al., 2014), we aim to enhance the efficiency of our analysis in the context of confabulations, which we will demonstrate empirically with EES and SeND.

## 2 INTERNAL TRAINING DYNAMICS

The training epochs of a transformer model can be vital in understanding the dynamics of how the model learns, particularly when trained on an unsupervised loss with stochastic gradient. Our analysis of training dynamics through multiple epochs in Appendix A shows that reducing stochastic gradient loss does not necessarily correspond to reducing hallucinations, verifying the results Li et al. (2024) showed for the oscillatory behaviour of LLMs in hallucination during training. Specifically, we found that increasing model size provides diminishing returns with respect to summarization in Figure 4b and has nearly no effect on self-consistency shown in Figure 4a. Most importantly, Figure 4a highlights the oscillatory hallucination behaviour throughout training.This highlights the need for further investigation into the relationship between optimization and factual accuracy in LLMs.

Following our investigation of the oscillatory behaviour in training, we look into the internal states of the Pythia 1B model to see what information we are able to extract. In doing so, we define a series of terms and formulas in order to understand the internal processes during the training of LLMs. This information is later used in sections 2.3 and 3 to assist us in deriving methods for improving the variance in the hallucinatory behaviour of models during training.

### 2.1 SENSITIVE NEURONS

To start our analysis of the internal states, we convert the activation matrix of the model into a sentence embedding vector 2.1 which turns an $\mathbb{R}^{n,m}$ activation matrix into a sentence embedding vector $a_k$ for input $k$ with dimension $\mathbb{R}^n$. Given its demonstrated success in hallucination detection by Su et al. (2024), we employ this sentence embedding extraction approach.

**Definition 2.1** (Sentence Embedding Vector). The Sentence Embedding Vector is a way to convert the large $\mathbb{R}^{n,m}$ activation matrix into a smaller, easier to manage vector with dimension $\mathbb{R}^n$.

$$e_k = \frac{1}{2}((\frac{1}{m}\sum_{i=1}^{m} H_{N-1}^i) + H_{N-1}^m) \tag{1}$$

Where $e_k$ is the activation of one input $k$, $m$ is the number of tokens in the sequence, and $N-1$ is the subtraction to get the penultimate layer index. The penultimate layer of the LLM, being the layer closest to the output probabilities, is our primary focus for hallucination analysis due to its rich information about output certainty.

Next, we define the Net Change Formula 2.2 as a way to extract information from the model indicative of oscillatory behaviour between checkpoints from the sentence embedding vector.

**Definition 2.2** (Net Change Formula). Let $e_i^t$ denote the embedding of data point $x$ at neuron $i$ of the contextual embedding after checkpoint/epoch $t$. Then we define the net change formula as

$$\Delta e_i^t = |e_i^t - e_i^{t-1}| \tag{2}$$

With these definitions, we can now describe the crux of our investigation: **Sensitive Neurons**. These Sensitive Neurons give us key parts of the model that we will prove contribute to the hallucination of LLM models. They can be used to adapt training procedures for lowering hallucination variation during training and better overall confidence at inference time. In essence, Sensitive Neurons are embedding indices in the sentence embedding from definition 2.1 that experience drastic changes between checkpoints/epochs of the training, something we believe is related to the oscillatory behaviour in hallucination performance. When finding the most Sensitive Neurons, we typically want to select the top $K\%$ neurons for a specific data point's representation. In our investigation we set $K = 20$.

**Definition 2.3** (Sensitive Neurons). Indices of the contextual embedding for data point $x$ which exhibit the highest net change across the last $C$ checkpoints of training, indicating overall high variability during this period. This is calculated by

$$V_i = Var(e_i) \sum_{t=T-C+1}^{T} \Delta e_i^t \tag{3}$$

where $V_i$ is the total variability during the last $C$ checkpoints and the most Sensitive Neurons are

$$\mathbf{s} = \arg \max_{1 \leq i \leq N} \{V_i \mid V_i \geq \text{percentile}(V, 100 - k)\} \tag{4}$$

where N is the embedding vector size and $k$ is the desired percentile threshold.

The above definition of Sensitive Neurons is then applied to LLM hallucinations through analyses of the EigenScores. In their paper, Chen et al. (2024) define a new metric for detecting confabulations, a subclass of hallucinations. They do this by calculating an EigenScore 2.4 based on determinant calculations from multiple outputs of an LLM with a high-temperature setting (*temperature* set to 0.5) to encourage the LLM to produce a variety of different outputs. They propose that if an LLM is set to hallucinate on that output, the generated texts will show higher semantic variability and produce a higher EigenScore. This method achieves SOTA performance and is unsupervised as it only relies on the representations learned by the model. In the forthcoming sections, we will analyze the correlation between the EigenScore of data points during training checkpoints and the most Sensitive Neurons associated with them.

**Definition 2.4** (EigenScore). The **EigenScore** of data point $x$ indicates the degree of hallucination on input $x$ by the average logarithm of the eigenvalues on the covariance matrix of the multiple output generations (typically 10 in our experiments).

$$ES = \mathbb{E}(Y \mid x, \theta) = \frac{1}{K} \sum_{i=1}^{K} \log(\lambda_i) \tag{5}$$

where $\lambda = \{\lambda_1, \ldots, \lambda_K\}$ denotes the eigenvalues of the regularized covariance matrix $\Sigma + \alpha \cdot \mathbb{I}$. we advise referring to Chen et al. (2024) for a more detailed analysis of this formula.

## 2.2 SENSITIVE NEURON IMPACT ON EIGENSCORES

To assess the correlation between Sensitive Neurons and other neurons in the embedding matrix of 10 generated outputs at a specific checkpoint, we conduct experiments aimed to determine if the presence of Sensitive Neurons indicates higher uncertainty and a greater likelihood of hallucinations.

We evaluate the Sensitive Neuron effect on the HELM dataset (Su et al., 2024), which includes outputs and internal states from six open-source LLMs based on inference over 50,000 Wikipedia articles, with human annotators labeling passages as factual or hallucinatory. This dataset was selected as Wikipedia is one of the main fact sources people refer to, and to reduce the spread of misinformation, LLMs should be robust to this type of information. To assess the impact of Sensitive Neurons on hallucination, we adapt the EigenScore method by applying it to sentence embeddings from the

penultimate layer of EleutherAI's Pythia 1B model, focusing on checkpoints between 133,000 and 143,000 training steps, where embeddings are more stable. We perform sensitive neuron dropout, removing the top 10% of Sensitive Neurons at each checkpoint, and compare the results to a baseline where 10% of neurons are randomly dropped. Additionally, we analyze the impact on hallucination-prone inputs versus non-hallucination-prone inputs to determine if Sensitive Neurons play a critical role during hallucination, without negatively affecting correct outputs.

### 2.2.1 WHAT IS THE EFFECT OF SENSITIVE NEURONS ON HALLUCINATION METRICS?

Since a reduction in the EigenScore metric can be used as a proxy to show the reduction in likelihood of hallucination, we keep using this metric in our investigations. We are able to show through our comparison of the baseline random neuron dropout and Sensitive Neuron dropout that Sensitive Neurons significantly reduce the EigenScore metric and in turn, reduce the possibility of a confabulation (Figure 1a). Not only do we observe this in hallucinatory outputs, we also observe a smaller reduction in EigenScore when applying this technique to correctly answered queries (Figure 1b). This result indicates that our methodology has a significant effect on the uncertainty shown by an LLM. We observe that looking at the internal states of the model is an effective way to eliminate confabulating text generation in various model sizes.

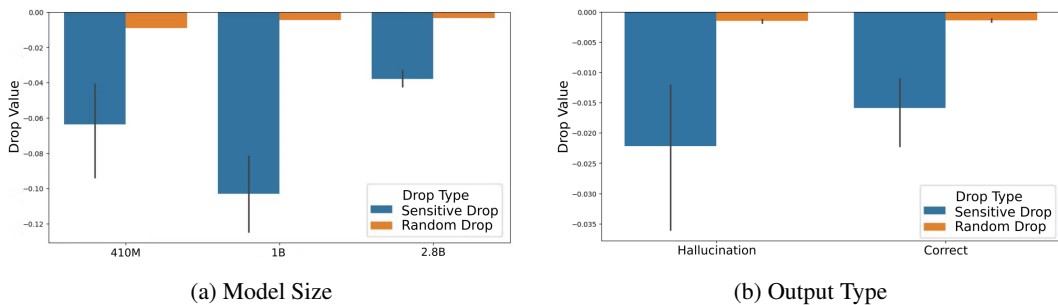

(a) Model Size          (b) Output Type

Figure 1: **Comparison of sensitive neuron dropout** on inference of Eleuther AI's Pythia various model sizes with random neuron dropout. (a) Average sensitive neuron dropout with standard deviation plotted as scale of the model increases. (b) Average sensitive neuron dropout for hallucinatory inputs and non-hallucinatory inputs. Input size for each test is 80 I.I.D. texts. Sensitive neuron dropping presents a clear, significant reduction in EigenScore compared to that of random neuron dropping across model sizes. Hallucinatory generations experience a larger drop in EigenScore, meaning that our protocol scales with likelihood of hallucination.

### 2.3 EFFICIENT EIGENSCORE APPROXIMATION

To address the computational complexity of EigenScore calculations, particularly as LLM hidden layer sizes increase, we develop an approximation method. This approximation, detailed in Algorithm 2, leverages the properties of Spectral Density or Density of States (DOS) to estimate EigenScore without explicitly constructing the covariance matrix. While this approximation provides a general overview of EigenScore trends, it is important to note that the output scales differ: EigenScore ranges from $[0, \infty)$, whereas the approximation, referred to as **Efficient EigenScore (EES)**, outputs values between $[-1, 1]$. Since the spectrum of the matrix is altered to make EES computable and operates on its own scale, EES can be seen as a standalone metric for hallucination detection.

The computation of the Efficient EigenScore (EES) is based on two fundamental concepts: Chebyshev Polynomials and Density of States (DOS). A detailed introduction to these concepts is provided in Appendix sections C.1 and C.2. Below, we outline a brief sketch of the derivation of EES. Since Chen et al. (2024) use the covariance matrix of the embedding matrix of 10 generated sequences by the model in their methods, we represent it with $H$ and use it in our derivation.

**Lemma 1.** *Let $f = \log$. Then, for a covariance matrix $H$ with eigenvalues $\lambda_i$, we have*

$$trace(\log(H)) = \sum_{i=1}^{N} \log(\lambda_i), \tag{6}$$

*where $\lambda_i$ are the eigenvalues of $H$.*

**Proposition 1.** *Using the property of the density of states (DOS), we have:*

$$\int \log(\lambda)\,\mu(\lambda)\,d\lambda = \log\left(\prod_{i=1}^{N}\lambda_i\right), \tag{7}$$

*which follows from Lemma 1 since $\sum_{i=1}^{N}\log(\lambda_i) = \log\left(\prod_{i=1}^{N}\lambda_i\right)$.*

Note that from Proposition 1, the integral is equal to $N.EigenScore(H)$ or in our application, given $C$ the integral equals $K.EigenScore(C)$, $K$ being the number of model generations.

Our objective is to simplify the integral and approximate its value, avoiding the direct computation of the covariance matrix. This approach is intended to mitigate the computational complexity and associated costs of explicitly handling the covariance matrix. Further utilizing Chebyshev Polynomials, DOS, and KPM (as introduced in Appendix C.2), we can simplify the integral mentioned in Equation 7 to $\sum_{m=0}^{M} d_m c_m$, where $d_m$ term in DOS is approximated using Stochastic Trace Estimation and $c_m$ m'th Chebyshev Polynomial coefficient. Appendices C.3 and C.4 provide the derivation of this equation. Note that the simplified integral is ultimately used to approximate the EigenScore of the matrix which is ultimately equivalent to $\frac{1}{K}\sum_{m=0}^{M} d_m c_m$. Performance of EES approximation is closely correlated with that of the original EigenScore and can be seen to closely track the progress of EigenScore through training of Pythia 1B on the HELM dataset in Figure 7.

## 2.4 HOW DOES EFFICIENT EIGENSCORE APPROXIMATION SCALE COMPARED TO REGULAR EIGENSCORE?

The efficiency of EES compared to regular EigenScore is evaluated for scaling matrix sizes, which is critical for applying our training protocol on LLMs (Section 3). We conduct a grid search over matrix size (Figure 2) and moments used in EES calculation (Figure 6). As shown in Figure 2, EES demonstrates a significant computational advantage, reducing computation time by nearly half for matrix sizes of $\mathbb{R}^{1e8}$, with EES taking around 4 seconds versus 7 seconds for EigenScore. Thus, EES offers substantial computational efficiency as model and matrix sizes increase.

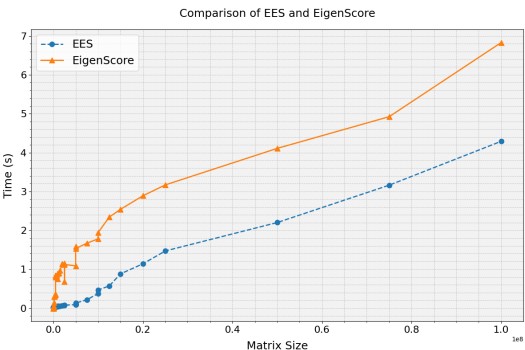

Figure 2: **Efficient EigenScore approximation scaling investigation**. The figure shows the difference in computation time between regular EigenScore calculation and EES with a moments value of 20. The x-axis represents the product of the matrix's rows and columns, and the y-axis shows the computation time. As matrix size increases, EES consistently reduces computation time, making it a practical choice for large LLMs.

## 3 SENSITIVE NEURON DROPOUT (SEND)

Building on the findings from Section 2.2, and aiming to reduce variance in the factual uncertainty of LLMs during training, this section introduces SeND, an efficient and transferable framework for training LLMs. SeND integrates the EES method discussed in Section 2.3 to enhance computational

efficiency while addressing variance in sensitive neuron behavior. By identifying sensitive neurons, which contribute to the oscillatory behavior of hallucinations during training, SeND deterministically drops these neurons based on a small subset of the training data. This approach ensures an increase in the model's factual certainty by the end of training as explained in Algorithm 1.

---

**Algorithm 1** Sensitive Neuron Dropout

---

**Require:** $\epsilon$ denotes the acceptable range for loss convergence and $\delta$ denotes acceptable range for confabulation (EES) convergence
  1: Initialize dataset with $\alpha\%$ training $Y_t$ and $(100 - \alpha)\%$ tracking $Y_s$
  2: **while** Loss $> \epsilon$ and EES $> \delta$ **do**                          ▷ Refer to Algorithm 2 for EES
  3:     **for** $t$ in T **do**          ▷ T denotes the number of epochs per sensitive neuron calculation
  4:         Train LLM for one epoch over $Y_t$
  5:         Record penultimate layer representations $R_t$ of LLM over $Y_s$
  6:     **end for**
  7:     **for** $t \in T - 1$ **do**
  8:         Calculate variability $V_t$ between $R_t$ to $R_{t+1}$             ▷ Refer to Equation 3
  9:     **end for**
10:     Take average Variability $V_{avg} = \frac{1}{N_s} \sum_{i=0}^{N_s} V_i$
11:     $s = K$ most sensitive neurons $\in V_{avg}$                 ▷ Refer to Equation 4
12:     Drop neurons $s$ for next T epochs
13: **end while**

---

### 3.1 SeND EXPERIMENT SETUP

To evaluate SeND, we use Eleuther AI's Pythia 1B model, continuing its training on specific datasets rather than restarting pretraining to maintain efficiency. We continue the training of the fully trained model on two datasets: HELM, consisting of Wikipedia text (Su et al., 2024), and MedHALT, a medical dataset emulating real-world entrance exam questions (Pal et al., 2023). Due to the importance of factual accuracy in the medical domain, MedHALT was chosen to assess SeND's impact on hallucination mitigation in an additional field where hallucinations are highly impactful. Both datasets were tested in two sizes: 200 and 2,000 points (referred to as 2k). SeND implements the EigenScore reduction technique from Section 2.2 and detects Sensitive Neurons using a 3-epoch window on a specialized hallucination tracking dataset. Sensitive Neurons in the penultimate layer are identified based on their variability across epochs and are deterministically dropped for the subsequent 3 training epochs. This dropout process is repeated at each 3-epoch interval until the training loss converges, effectively mitigating hallucination tendencies and refining the model.

### 3.2 PERFORMANCE OF SeND ON PYTHIA 1B

The results of running Pythia 1B on HELM and MedHALT 200 are illustrated in Figure 3. To validate that the EES method accurately approximates the EigenScore metric; we compare the model's progress during training (up to loss convergence) and assess whether the resulting graphs are similar. These results are detailed in Appendix C.7. Upon confirming that EES provides a reliable approximation of the EigenScore, we proceed to compare the performance of Pythia 1B trained using standard training without dropout to that of Pythia 1B trained with SeND on HELM and MedHALT 2k (Figure 3). A baseline of no dropout was used for comparison as experiments showed that implementing random dropout resulted in worse performance. In the case of training on HELM with the regular protocol, we observe results consistent with previous findings: while the model successfully reduces loss, it fails to optimize for hallucination, as evidenced by the increasing EES metric (green line in Figure 3a). Conversely, training with SeND reveals a consistent trajectory toward reducing both EES and loss, as depicted by the blue line.

To assess the effectiveness of SeND in comparison to other state-of-the-art factuality metrics, we employ the FactScore metric from Min et al. (2023), which quantifies the factual accuracy of content generated by large language models (LLMs). The fact-checking is conducted using the HELM dataset where a higher FactScore indicates improved factual precision. When evaluated on 100 data points from the HELM dataset, the 1B SeND model achieves a FactScore of 0.07, whereas the 1B Normal Training model attains 0.05, demonstrating a 40% improvement in factual accuracy, even

during test time. This highlights the efficacy of SeND in enhancing the factual certainty of the model. Note that SeND is not a replacement for post-hoc methods such as RAG (Gao et al., 2024b), but rather to complement them.

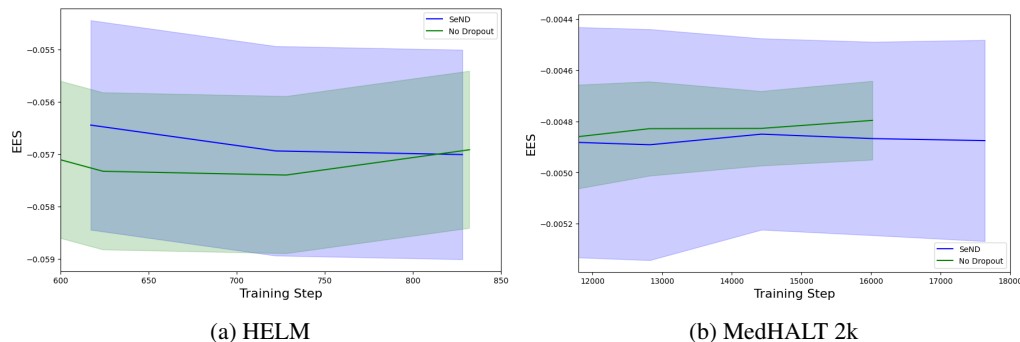

(a) HELM          (b) MedHALT 2k

Figure 3: **Regular finetuning vs. SeND on two datasets**. (a) presents the results of training Pythia 1B on HELM with regular finetuning and SeND. (b) uses the same training setup as (a), but the LLM is trained on MedHALT 2k. In both plots, performance is reported as the average **EES** over 5 runs on the validation set. Models are trained until loss convergence. Training with SeND shows a more controlled reduction in **EES** compared to regular finetuning, suggesting that SeND optimizes for hallucinations as well as loss, with less overall confidence variability during training.

The robustness of this training protocol is essential as we aim for it to be applied across many different fields. In light of this, we present the results of finetuning on MedHALT 2k with and without SeND in Figure 3b. We observe a similar trend in 3b as shown in Figure 3a, where standard finetuning increases the EES score throughout training, showing that the model is not taking into account hallucinations and factuality during its training. In Figure 3b, there is an improvement in the trajectory of EES as training continues, showing that our model is in fact able to incorporate factuality as a metric to account for during training of the model. The small difference observed between the training protocol behaviours could be due to MedHALT 2k data never being seen before the finetuning phase whereas HELM data has been seen. In this case, it may be beneficial to delay the onset of SeND, as high variability between checkpoints on new training instances is expected.

## 4   CONCLUSION & FUTURE WORK

In this paper, we presented a protocol to refine the current training methods of LLMs based on experiments showing oscillatory behaviour with respect to hallucinations throughout training (Figure 4). To do this we used the internal states of LLMs, specifically the penultimate layer activations during inference on a specialized dataset. We present an initial method of reducing hallucinations based on the principles of EigenScore metrics introduced by Chen et al. (2024). We showed empirically that our Sensitive Neuron detection method significantly reduces the EigenScore on inference of LLMs throughout various stages of training (Figure 1). Following the success of the Sensitive Neuron method, we moved on to the application of a hallucination reduction method on training. We show through finetuning that we are able to fix the oscillatory behaviour initially seen throughout training and reduce the EES of finetuned models as shown in Figure 3 by modifying the internal mechanics of training with **SEnsitive Neuron Dropout**. At test time, in conjunction with Retrival Augmented Generation, we achieve a 40% increase in FactScore performance, verifying that SeND provides a substantial improvement to current training protocols.

In the future, we would like to scale our method to larger datasets and larger models as we faced compute power restrictions with larger LLMs. Proving the performance of SeND on larger open source models such as Meta's Llama 3.1 (Dubey et al., 2024) will give organizations creating state of the art LLMs the evidence they need to implement SeND into their training protocol and launch safer models. We also expect SeND to perform even better on larger LLMs since we are introducing a regularization technique to reduce variance during training. Applying this to larger LLMs with an innate higher variance could see SeND having a larger impact on the model.

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

## A  OSCILLATORY BEHAVIOUR VALIDATION

### A.1  METHODS

In our study, we utilize Eleuther AI's Pythia and LMEval tools (Biderman et al., 2023; Gao et al., 2024a), to examine the development and evolution of LLMs throughout the training process. Pythia comprises a suite of 16 LLMs, all trained on public data in the same sequential order, with sizes ranging from 70 million to 12 billion parameters, 8 of which we use for our experiments. We used 20 equally spaced training checkpoints from the start to the finish for our analysis. We chose Pythia as it is based on GPT-Neo X (Black et al., 2022), which shares a similar foundational architecture to other state-of-the-art language models. Pythia's comprehensive package of models makes it particularly suitable for our analysis, allowing us to conduct a thorough examination of the development and evolution of LLMs throughout the training process.

These models are evaluated at each checkpoint on a variety of hallucination/fact-checking metrics. To do this, we leverage the HuggingFace Hallucination Leaderboard (Hong et al., 2024), which offers comprehensive benchmarks for our experiments. There are two main components to our

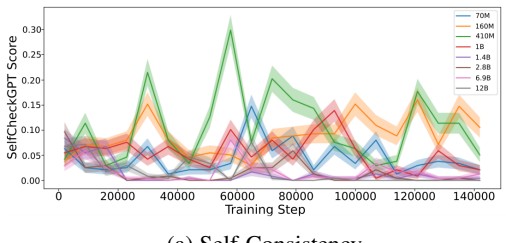
(a) Self-Consistency

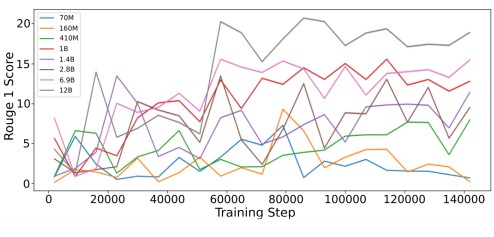
(b) Summarization

Figure 4: **Visualization of Oscillatory Behavior Across Varying LLM Sizes**. Hallucination metrics are evaluated at equidistant checkpoints of the Pythia models, with sizes 70M, 160M, 410M, 1B, 1.4B, 2.8B, 6.9B, 12B. Part (a) presents the performance of the Pythia models under the SelfCheck-GPT metric. Average performance is indicated by solid lines, while the shaded regions represent the standard deviation. Higher SelfCheckGPT score indicates a higher probability of self-contradiction. Part (b) depicts the same experimental setup, but hallucination measured on the XSum v2 dataset, where Rouge1 is used as the performance metric. A higher Rouge1 score suggests a better alignment of the generated text to that of the reference summary. For all model sizes, we observe a pronounced trend of high variance and oscillatory behavior in hallucination rates, highlighting the model's uncertainty and need for a robust mitigation strategy to stabilize performance.

evaluations: Summarization and Self-consistency. For summarization, models are evaluated under the XSum dataset Narayan et al. (2018) where the model is given a dataset of BBC news articles and must give summaries of each article. A higher Rouge1 score on XSum means the data is aligning better with the provided reference summary. SelfCheckGPT (Manakul et al., 2023) is used to see if the model is uncertain with respect to a dataset of prompts for self-consistency. When the SelfCheckGPT score is high, it means the model is more likely to contradict itself on the given input and therefore, more likely to hallucinate.

## A.2 How do the established iterative training processes influence LLM hallucinations?

The analysis of hallucination oscillations, as shown in Figure 4, indicates a consistent pattern across different models: oscillations persist throughout training from the initial to the final checkpoint. This finding highlights the uncertainty of halting training solely based on the convergence of training loss to its minimum to minimize hallucination. For example, in the evaluation plot of XSUM with the 12B model (4b), the optimal value for the hallucination metric occurs at the much before training termination. This evidence challenges the notion that optimizing solely for unsupervised loss in SGD guarantees learning the most accurate representation of the data. This observation is seen more drastically in 4a, where model size has even less effect on the performance of SelfCheckGPT. Instead, we observe exaggerated oscillatory behaviour within self-consistency, meaning that model size is even less effective at tackling the issue of confabulations. One potential solution to mitigate this issue could involve incorporating a regularization term into the unsupervised loss based on a hallucination detection metric discussed in Section 3.

## A.3 How does model complexity affect the emergence of hallucinations throughout training?

An analysis of hallucination detection metrics reveals a diminishing rate of improvement with increased model scaling, particularly up to the 12B parameter size (Figure 4b). This suggests that beyond a certain point, even though there is improvement in the hallucinations, larger models do not significantly reduce hallucinations, indicating that scaling alone is not sufficient for building robust models. Instead, more refined approaches are needed to address the underlying variability in model behavior. For the following experiments, we focus on the Pythia 1B model.

## B  ADDITIONAL EXPERIMENTS

### B.1  DRASTIC EMBEDDING CHANGES LEADING TO SENSITIVE NEURONS

Looking at internal states of the model allows getting a deeper understanding of the dynamics that could be leading to the oscillatory behaviour seen in Figure 4. To do this, we record the net change (Definition 2.2) between checkpoints of the penultimate layer where one checkpoint would be the correct answer and the next would hallucinate. This net change with respect to various different input texts is plotted in Figure 5. It can be observed that there were specific embedding activations that experienced drastically more change relative to the rest of the embeddings. This is the main source of motivation to further define Sensitive Neurons (Definition 2.3).

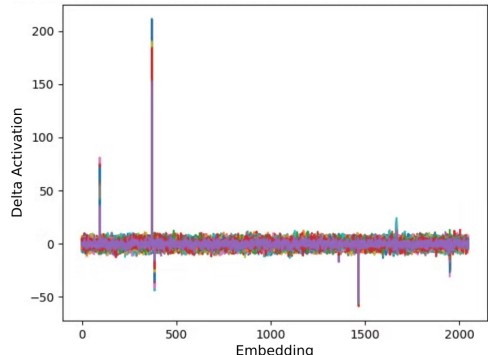

Figure 5: **Net change of sentence embeddings** between checkpoints 125,000 and 143,000. Each different colour is a different input text. As depicted, there are specific neurons that go through drastic changes between the two checkpoints of the training regardless of the input.

## C  EFFICIENT EIGENSCORE (EES) DERIVATION

### C.1  BACKGROUND: CHEBYSHEV POLYNOMIALS

Chebyshev polynomials are a sequence of orthogonal polynomials in the interval $[-1, 1]$ – orthogonality property shown in equation 8 – that are widely used in numerical analysis, approximation theory, and other areas of applied mathematics. In this work, we are mainly concerned with the Chebyshev polynomials of the first kind with the recurrence relation shown in equation 9. Note that this recurrence could also be applied to matrices. Any function $f$ defined in the interval $[-1, 1]$ can be approximated with the Chebyshev expansion as shown in 10.

$$\int_{-1}^{1} \frac{2}{(1 + \delta_{0n})\pi\sqrt{1 - x^2}} T_m(x)T_n(x)\, dx = \delta_{mn},$$

$$\text{where} \quad \delta_{mn} = \begin{cases} 1 & \text{if } m = n, \\ 0 & \text{if } m \neq n, \end{cases} \tag{8}$$

$$
\begin{aligned}
T_0(x) &= 1, \\
T_1(x) &= x, \\
T_{n+1}(x) &= 2x \cdot T_n(x) - T_{n-1}(x), \quad \text{for } n \geq 1.
\end{aligned}
\tag{9}
$$

$$f(x) = \sum_{n=0}^{\infty} c_n T_n(x), \tag{10}$$

$$\text{where } c_n = \frac{2}{\pi} \int_{-1}^{1} \frac{f(x)T_n(x)}{\sqrt{1-x^2}} \, dx \text{ for } n > 0, \tag{11}$$

$$c_0 = \frac{1}{\pi} \int_{-1}^{1} \frac{f(x)}{\sqrt{1-x^2}} \, dx. \tag{12}$$

## C.2 BACKGROUND: DOS AND KPM

Let $H$ be a symmetric matrix $H \in \mathbb{R}^{N \times N}$ with an eigendecomposition $H = Q\Lambda Q^T$, where $\Lambda = \text{diag}(\lambda_1, \cdots, \lambda_N)$ and $Q = [q_1, \cdots, q_N]$ is orthogonal. The spectral density induced by $H$ is the generalized function:

$$\mu(\lambda) = \frac{1}{N} \sum_{i=1}^{N} \delta(\lambda - \lambda_i), \tag{13}$$

where $\delta$ is the Dirac delta function. For any analytic test function $f$, the integral of $f$ with respect to $\mu$ is:

$$\int f(\lambda)\mu(\lambda) \, d\lambda = \text{trace}(f(H)). \tag{14}$$

Dong et al. (2019) introduced KPM as a numerical technique to approximate DOS. KPM approximates DOS by expanding it in terms of chebyshev polynomials. Requiring the matrix's spectrum to be supported in the interval $[-1, 1]$, KPM approximates DOS with the following formula, $\lambda$ being the eigen value of the matrix $H$ and $d_m$ approximated by Stochastic Trace Estimation:

$$\mu^{\approx}(\lambda) = \sum_{m=1}^{\infty} d_m T_m^*(\lambda), \tag{15}$$

$$\text{where} \quad d_m = \frac{1}{N}\text{trace}(T_m(H)), \tag{16}$$

$$\text{and} \quad d_m \approx \frac{1}{N} \frac{1}{N_z} \sum_{j=1}^{N_z} \mathbf{z}_j^T T_m(H)\mathbf{z}_j, \tag{17}$$

$$\text{and} \quad T_m^*(x) = \frac{2}{(1 + \delta_{0m})\pi\sqrt{1-x^2}} T_m(x). \tag{18}$$

In the application for hallucination detection, we can use equation 14 to derive a formula for the EigenScore approximation using the properties of Chebyshev polynomials and DOS.

## C.3 STOCHASTIC TRACE ESTIMATION ON EMBEDDING MATRIX

We are interested in computing the $d_m$ term of DOS relying solely on the embedding matrix $E$ therefore we need to rewrite $d_m$ as follows:

$$d_m = \frac{1}{K} \frac{1}{N_z} \sum_{j=0}^{\infty} z_j^T T_m(E^T E)z_j \tag{19}$$

where $T_m$ can be computed using the Chebyshev polynomials of matrix $C = E^T E$.

$$T_0(E^T E)\mathbf{z}_j = I\mathbf{z}_j = \mathbf{z}_j,$$
$$T_1(E^T E)\mathbf{z}_j = E^T E\mathbf{z}_j,$$
$$T_{m+1}(E^T E)\mathbf{z}_j = 2E^T E T_m(E^T E)\mathbf{z}_j - T_{m-1}(E^T E)\mathbf{z}_j$$

Each term can be computed with a matrix-vector multiplication.

### C.4 EES INTEGRAL CALCULATION

Given the orthogonality of the Chebyshev polynomials, we can simplify the integral mentioned in proposition 1. To approximate the EigenScore, we will expand $\log(\lambda)$ in terms of Chebyshev polynomials and use their orthogonality to simplify the integral.

**Expanding and Integrating**

To approximate the integral:

$$\frac{1}{K} \int \log(\lambda)\mu(\lambda)\, d\lambda \tag{20}$$

Substitute the Chebyshev Expansion for DOS:

$$\mu(\lambda) \approx \sum_{m=0}^{M} d_m T_m^*(\lambda) \tag{21}$$

where:

$$T_m^*(\lambda) = w(\lambda)T_m(\lambda) = \frac{2}{\pi\sqrt{1-\lambda^2}(1+\delta_{0m})}T_m(\lambda)$$

Distribute $\log(\lambda)$ in the integral:

$$\frac{1}{K} \int \log(\lambda)\left(\sum_{m=0}^{M} d_m T_m^*(\lambda)\right) d\lambda = \frac{1}{K}\sum_{m=0}^{M} d_m \int \log(\lambda)T_m^*(\lambda)\, d\lambda \tag{22}$$

**Evaluate the Integral Using Orthogonality:**

To simplify the integral,

$$\int \log(\lambda)T_m^*(\lambda)\, d\lambda \tag{23}$$

First, express $\log(\lambda)$ as a series of Chebyshev polynomials:

$$\log(\lambda) = \sum_{m=0}^{\infty} c_m T_m(\lambda) \tag{24}$$

Then:

$$\int_0^1 \log(\lambda)T_m^*(\lambda)\, d\lambda = \int_0^1 \left(\sum_{m=0}^{\infty} c_m T_m(\lambda)\right) T_m(\lambda)\, d\lambda \tag{25}$$

$$\tag{26}$$

Note: The lower bound of the integral is 0 as the matrix is defined in the spectrum $[0, 1]$.

Using the orthogonality, we get:

$$c_m = \int_0^1 \log(\lambda) T_m^*(\lambda) \, d\lambda \tag{27}$$

So the integral simplifies to:

$$\frac{1}{K} \sum_{m=0}^M d_m c_m \tag{28}$$

## C.5 EFFICIENT EIGENSCORE ALGORITHM

Here we present a step by step guide on how to integrate the above derivations into a computation algorithm for the approximation of EigenScore.

---

**Algorithm 2** Efficient EigenScore (EES) Computation Algorithm

---

**Require:** Embedding matrix $E \in \mathbb{R}^{d_{\text{model}} \times K}$, number of Chebyshev terms $M$, number of stochastic trace estimation samples $N_z$

**Ensure:** Approximated EigenScore *EES*

1: **Standardize and Scale the Embedding Matrix $E$:**
2: $\quad E_{\text{mean}} = \frac{1}{K} \sum_{i=1}^K E[:, i]$            $\triangleright$ Compute mean of $E$
3: $\quad E_{\text{std}} = \sqrt{\frac{1}{K} \sum_{i=1}^K (E[:, i] - E_{\text{mean}})^2}$       $\triangleright$ Compute standard deviation of $E$
4: $\quad E_{\text{normalized}} = \frac{E - E_{\text{mean}}}{E_{\text{std}}}$          $\triangleright$ Standardize $E$
5: $\quad \sigma_{\text{max}} = \text{Power Method}(E_{\text{normalized}})$      $\triangleright$ Compute the largest singular value using the power method
6: $\quad E_{\text{normalized}} \leftarrow \frac{E_{\text{normalized}}}{\sigma_{\text{max}}}$          $\triangleright$ Scale $E$ by $\sigma_{\text{max}}$
7: **Initialize:**
8: $\quad d_m = 0 \quad \forall m \in \{0, 1, \ldots, M\}$        $\triangleright$ Initialize $d_m$ coefficients
9: $\quad c_m = 0 \quad \forall m \in \{0, 1, \ldots, M\}$        $\triangleright$ Initialize $c_m$ coefficients
10: **Compute DOS coefficients $d_m$:**
11: **for** $m = 0$ to $M$ **do**
12:      **Sample** $z_j \sim \mathcal{N}(0, I)$      $\triangleright$ Sample random vectors for stochastic trace estimation
13:      **Compute Chebyshev polynomial using the recurrence relation**
14: **end for**
15: **Compute Chebyshev coefficients $c_m$:**
16: **for** $m = 0$ to $M$ **do**
17:      $c_m \leftarrow \int_0^1 \log(\lambda) T_m^*(\lambda) \, d\lambda$      $\triangleright$ Using Equation 27 and Gaussian Quadrature for approximation
18: **end for**
19: **Compute EigenScore:**
20: $EES \leftarrow \frac{1}{K} \sum_{m=0}^M d_m c_m$      $\triangleright$ Approximate EigenScore using DOS coefficients
21: **return** $EES$      $\triangleright$ Return the approximated EigenScore

---

## C.6 EFFICIENT EIGENSCORE MOMENTS

Figure 6 presents the effect of using different moment values as the number of matrix rows increases with respect to time. This is an important hyperparameter to tune as increasing the number of moments on EES correlates to having a more accurate and representative approximation of the EigenScore. We observe that as moments in EES increase, the time to calculate EES increases. From this result, we conclude that selecting a moment value of under 50 would provide a balanced trade-off between accuracy and calculation time.

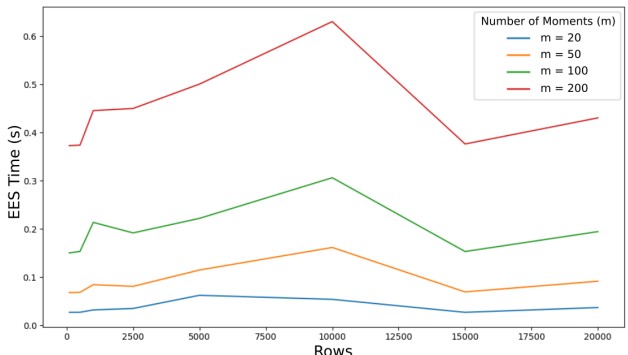

Figure 6:
Effect of changing number of moments on EES calculation time (seconds). More moments gives more accurate approximation but higher computation time.

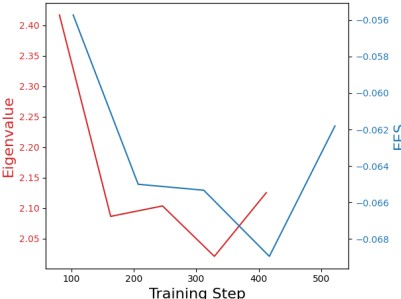

Figure 7: Performance of SeND on Pythia 1B wih HELM dataset computed with both EES and regular EigenScore. EES is able to closely track the true EigenScore performance metric, showing that it is a good approximator.

## C.7 EIGENSCORE AND EES TRAINING TRAJECTORIES

To demonstrate that our EigenScore approximation method, EES, is a good metric, we record the progress of Pythia 1B finetuning on the HELM dataset using both EigenScore and EES hallucination performance metrics (Figure 7). Albeit a different scale and window, the trajectories, magnitude and shape of the graphs are nearly identical while EES takes only 4 minutes to calculate and Eigen-Score takes approximately 8, an astounding 2x increase in compute speed. These results show that our metric closely resembles the target metric while greatly reducing the required computational resources.

