# OpenReview forum: "Hallucination Detox: Sensitive Neuron Dropout (SeND) for Large Language Model Training"
_NeurIPS.cc/2024/Workshop/SafeGenAi — SafeGenAi Poster_

### Official Review · Reviewer_KXSP · 2024-10-08
**The paper aligns well with the workshop theme and presents a novel approach with Sensitive Neuron Dropout (SeND), introducing an efficient way to compute EigenScore. However, the writing is unclear, with critical results in the appendix and some questionable methodological comparisons, and it lacks analysis on neuron sensitivity across epochs. These issues lead me to rate the paper below the acceptance threshold.**

**Rating:** 5
**Confidence:** 4

**Review:**

**Weakness**

_Writing style and Clarity:_

* Overall the writing style is poor
* Thought the paper is 8 pages long, it did not use the space efficiently. In section "3.2 PERFORMANCE OF SEND ON PYTHIA 1B" they are pointing to important results in Appendix.
* It is not very clear, why authors are adding last toke matrix in equation 1

_Missing analysis:_

* It is not very clear why authors compared SEND and no dropout in Figure 3. While the sensible approach would be to compare SEND with random dropout.
* Sensitivity of neurons is defined as change in activation between epochs and SEND drops the neurons with high sensitivity. The problem here is that activations change a lot during the initial epochs and change very less in the later epochs. The papers does not show any observations regarding this.


---

**Strengths**

_Alignment with the workshop:_

The paper aligns really with the theme of the workshop. The paper proposes Sensitive Neuron Dropout (SeND) where the neurons in the penultimate layer with high variation b/w epochs are dropped. The papers also proposed an efficient way to compute EigenScore called EES(Efficient EigenScore)

_Originality and Significance:_

The paper is proposing a novel approach of dropping neurons with high variance (sensitive neurons) which was not explored before.

---

### Official Review · Reviewer_8eVa · 2024-10-09
**Hallucination Detox: Sensitive Neuron Dropout (SeND) for Large Language Model Training**

**Rating:** 8
**Confidence:** 4

**Review:**

Summary:

This paper introduces a new method, SEnsitive Neuron Dropout (SeND), a training protocol created to mitigate hallucinations, by reducing variance during training. SeND systematically drops neurons with significant variability, also known as Sensitive Neurons, which show oscillations in contextual embeddings throughout training. Their work is focused on a specific type of hallucination, confabulations, where the model will generate inconsistent responses to inputs of a similar kind. This work is motivated by their empirical verification of the oscillatory nature of hallucinations when LLMs are trained.

Additionally, they introduced an Efficient EigenScore (EES), a hallucination detection metric, which approximates the traditional EigenScore at twice the speed. They claim when an LLM hallucinates, its generated texts will possess higher semantic variability and thus produce a higher EigenScore.

They ran experiments using Eleuther AI’s Pythia 1B model. They used two datasets,  HELM, consisting of Wikipedia text, and MedHALT, a medical dataset with synthetic entrance exam questions. When applied in addition with Retrieval Augmented Generation, they achieved a 40% increase in FactScore performance, illustrating that SeND provides a significant improvement to training protocols.

Pros:
The method focuses on hallucinations that emerge during the training process, while previous research has focused primarily on post hoc detection and mitigation strategies.
Applied their method in addition with previous hallucination-mitigation methods such as Retrieval Augmented Generation, SeND is a complement to post-hoc methods.
Research expanded to the medical domain, using the MedHALT dataset to evaluate SeND’s impact on hallucination mitigation.
Compared sensitive neuron dropping to random neuron dropping, saw a significant reduction of EigenScore.

---

### Official Review · Reviewer_p6Pu · 2024-10-10
**The paper presents the Sensitive Neuron Dropout (SeND) training protocol as a novel and effective approach to mitigate hallucinations in Large Language Models, however it would benefit from broader model and dataset evaluations to prove scalability of their methods..**

**Rating:** 8
**Confidence:** 5

**Review:**

### Strengths:
- **Novelty of Method and Benchmark**: Introduces a new training protocol to mitigate hallucinations during training of LLMs to increase factual confidence during training.
- **Problem Definition**: By identifying how hallucinations fluctuate during training, the authors lay clear grounds to their newly proposed method.
- **Strong performance**: A simple but efficient (with EES score) method that improves LLM reliability at test time by up to 40% compared to normal training.

### Weaknesses:
- **Limited Model Scope**: The study primarily evaluates SeND on the Pythia 1B model and does not explore its effectiveness across a broader range of LLM architectures.
- **Choice of datasets**: The choice of HELM and MedHALT for training should be addressed. Also to show robustness of the method across different datasets, authors should include different types.
- Targeting hallucinatory neurons during training is an existing technique for hallucination mitigation and some novelty of the paper lies in the EES score.The authors need to show that this is scalable in larger datasets and larger models. The experiments lack diversity in size and type of models.